# Paternally expressed gene 3 (*Pw1/Peg3*) promotes sexual dimorphism in metabolism and behavior

Karo Tanaka[1], Vanessa Besson [1], Manon Rivagorda[2¤a], Franck Oury [2¤a], Giovanna Marazzi[1¤b], David A. Sassoon [1¤b]*

1 Stem Cells and Regenerative Medicine, Institute of Cardiometabolism and Nutrition (ICAN), INSERM U1166, University of Pierre and Marie Curie Paris VI, Paris, France, 2 Hormonal Regulation of Brain Development and Functions, INSERM U1151, Institut Necker Enfants Malades, Paris, France

¤a Current address: Integrative neurobiology, INSERM U1151, Institut Necker Enfants Malades, Paris, France
¤b Current address: Paris Cardiovascular Research Center, INSERM U970, European Hospital Georges Pompidou, Paris, France
* david.a.sassoon@gmail.com

**Data Availability Statement:** All relevant data are within the manuscript and its Supporting Information files.

## Abstract

The *paternally expressed gene 3* (*Pw1/Peg3*) is a mammalian-specific parentally imprinted gene expressed in stem/progenitor cells of the brain and endocrine tissues. Here, we compared phenotypic characteristics in *Pw1/Peg3* deficient male and female mice. Our findings indicate that *Pw1/Peg3* is a key player for the determination of sexual dimorphism in metabolism and behavior. Mice carrying a paternally inherited *Pw1/Peg3* mutant allele manifested postnatal deficits in GH/IGF dependent growth before weaning, sex steroid dependent masculinization during puberty, and insulin dependent fat accumulation in adulthood. As a result, *Pw1/Peg3* deficient mice develop a sex-dependent global shift of body metabolism towards accelerated adiposity, diabetic-like insulin resistance, and fatty liver. Furthermore, *Pw1/Peg3* deficient males displayed reduced social dominance and competitiveness concomitant with alterations in the vasopressinergic architecture in the brain. This study demonstrates that *Pw1/Peg3* provides an epigenetic context that promotes male-specific characteristics through sex steroid pathways during postnatal development.

## Author summary

*Pw1/Peg3* is under parental specific epigenetic regulation. We propose that *Pw1/Peg3* confers a selective advantage in mammals by regulating sexual dimorphism. To address this question, we examined the consequences of *Pw1/Peg3* loss of function in mice in an age- and sex-dependent context and found that *Pw1/Peg3* mutants display reduced sexual dimorphism in growth, metabolism and behaviors. Our findings support the intralocus sexual conflict model of genomic imprinting where it contributes in sexual differentiation. Furthermore, our observations provide a unifying role of sex steroid signaling as a

**Funding:** This work was supported by the French Ministry of Research Chaire d'Excellence and the European Community Seventh Framework Program projects ENDOSTEM (Activation of vasculature associated stem cells and muscle stem cells for the repair and maintenance of muscle tissue-agreement number 241440) and support from the Agence Nationale de la Recherche (Laboratoire d'Excellence Revive, Investissement d'Avenir; ANR-10-LABX-73) and Carmaa (RHU-ANR). We also thank Inserm and the University of Paris (VI) Sorbonne for institutional support. The funders had no role in study design, data collection and analysis, decision to publish, or preparation of the manuscript.

**Competing interests:** The authors have declared that no competing interests exist.

common property of *Pw1/Peg3* expressing stem/progenitor cells and differentiated endocrine cells, both of which remain proliferative in response to gonadal hormones in adult life.

## Introduction

Parental genomic imprinting is a form of epigenetic regulation by which one allele of a gene is expressed according to its parent-of-origin. In vertebrates, this form of imprinting is unique to placental mammals and its evolutionary advantage is still under active debate [1–3]. The parental conflict (or kinship) [4] and maternal-offspring coadaptation theories [5] are two widely recognized concepts to explain why parental genomic imprinting arose in mammals. Independently, Day and Bonduriansky proposed an 'intralocus sexual conflict' model [6] that predicts a physiological role for genomic imprinting in the genetic architecture of sexually dimorphic traits. This hypothesis is applicable to any species and traits under sex-specific selection pressure. However, empirical exploration of the role of imprinted genes in sexual differentiation is relatively limited [7,8].

Human diseases associated with deregulated genomic imprinting and gene knockout studies in mice have established pivotal roles of genomic imprinting in growth, metabolism, reproduction, and behavior [9–11]. In human and mouse, many imprinted genes are clustered in distinct chromosomal regions and are typically co-expressed in organs and tissues that regulate homeostasis of the whole-body energy metabolism, such as the brain hypothalamic region, liver, pancreas, muscle, fat, gonads, and placenta [9,12,13]. The generation of mutants corresponding to several imprinted genes in mice demonstrated global metabolic changes and their imprinting status (i.e. maternal or paternal) often correlates with inverse metabolic outcomes. Specifically, paternally expressed genes such as *Magel2*, *Dlk1* and *Zac1* promote growth and energy expenditure and restrict adiposity whereas the maternally expressed genes, *H19* and *Grb10*, suppress growth and increase adiposity ([10] and references therein). Genome-wide transcriptome analyses have further demonstrated that inactivation or overexpression of a single imprinted gene alters the expression profile of multiple imprinted genes, suggesting that imprinted genes act in networks to coordinate cellular and organ development and functions [14].

The *Pw1/paternally expressed gene 3* (hereafter referred to as *Pw1)* is a mammalian-specific, parentally imprinted gene that is widely expressed during early embryonic development and becomes restricted to subset of tissues in adulthood [15,16]. Using a *Pw1* reporter transgenic mouse line (Pw1[IRESnLacZ]), we showed that *Pw1* is expressed in a wide array of adult stem/progenitor cells [17]. Studies of different types of progenitor cells, all of which express high levels of *Pw1*, demonstrated that *Pw1* dysfunction alters stem cell competence, self-renewal capacity, and cell cycle behaviors [18–21]. At a molecular level, *Pw1* modulates cell stress pathways including TNFα-NFκB signaling in cell growth and survival [22], p53 signaling in apoptosis [23,24], and decolin-induced autophagy [25]. The PW1 protein also acts as a transcription factor that is shown to regulate expression of mitochondrial genes in the brain [26] as well as oxytocin receptor [27]. To date, several lines of *Pw1* mutant mice have been generated by different gene targeting strategies [28–30]. Mice carrying a paternally inherited mutant allele for *Pw1* consistently displayed reduced pre- and postnatal growth in all models. Pw1[+/p-] adult males were also shown to have altered energy homeostasis such as increased body fat and reduced thermogenesis, whereas metabolic phenotypes of female counterparts were not fully characterized in detail [31]. By contrast, a delayed onset of oestrus cycle and alterations in the reproductive physiology, such as smaller litter size and mature oocytes, were demonstrated [30,31]. It has been also reported that *Pw1* deficient mice display deficits in adaptive traits, such as

maternal care in females [28], and sexual experience-dependent olfactory learning in males [32]. All these findings indicate a significant involvement of *Pw1* in sex-hormone dependent physiology, but the underlying mechanism by which this paternally expressed gene exerts such diverse biological functions remained unresolved.

In this study, we characterized paternally inherited *Pw1* deficient phenotypes in male and female mice at different stages of postnatal development. We identified specific growth factor and hormonal axes that are deregulated at critical stages of postnatal development. At the cellular level, we demonstrate co-localization of *Pw1* in sex-hormone dependent cell types in various organs. Our results point to a central role for *Pw1* in establishing sexual dimorphism in mammals that regulates overall sex-specific physical traits, metabolism and behavior.

## Results

### Reduced masculinization of growth and metabolism in *Pw1* deficient males

Mice carrying a paternally inherited mutant allele ($Pw1^{+/pat}$) were distinguishable from their wildtype (WT) littermates ($Pw1^{+/+}$) by a smaller size at birth and a reduced postnatal weight gain, as previously reported by our group and others [19,28,29]. Comparison of male and female littermates in the postnatal growth phase revealed that body weight was identical between males and females up to 3.5 weeks of age in both genotypes and $Pw1^{+/p-}$ mice were significantly smaller (Fig 1A). Sex differences emerge at 4 weeks of age in both genotypes, with slight delay in the $Pw1^{+/pat}$ littermates. Body weight at 7 weeks of age revealed a significant interaction between sex and genotype (p<0.05), and $Pw1^{+/pat-}$ are significantly smaller in both sexes (p<0.0001) while females are significantly smaller than males (p<0.0001). Multiple comparisons revealed all four groups are different (p<0.0001), however notably, there were no differences detected between $Pw1^{+/+}$ males and $Pw1^{+/pat-}$ females. We observed a positive correlation between random-fed blood glucose and body weight during the postnatal growth phase regardless of sex and genotype (Fig 1B, left), and the $Pw1^{+/pat-}$ mice displayed reduced glucose levels up to 2 months of age (Fig 1B, right, p<0.001). Concomitantly, food consumption between 2 months and 2.5 months of age was reduced in a sex dependent manner (Fig 1C). We noted that for all sexually dimorphic parameters examined, $Pw1^{+/pat-}$ males were similar to $Pw1^{+/+}$ female littermates, which indicated a role for *Pw1* in the control of male-specific sexual differentiation during postnatal development.

In adult mammals, including humans and mice, males are typically larger with an increased skeletal mass, whereas females are smaller with higher adiposity. To further monitor the sex-dependent postnatal development and maturation, we performed a longitudinal analysis of body composition (lean/fat mass) of $Pw1^{+/pat-}$ males and females in comparison to $Pw1^{+/+}$ littermates using non-invasive NMR imaging. During secondary sexual maturation, both male and female $Pw1^{+/pat-}$ animals showed reduced lean mass at all time points analyzed, but the difference became more marked in males (by 20%, p<0.001) than in females (by 15%, p<0.001) (Fig 1D, top left). In contrast, the fat mass development was highly sex-dependent. $Pw1^{+/+}$ males manifested transient reduction of fat mass at 3 months of age whereas $Pw1^{+/pat-}$ males did not undergo this transition resulting in an accelerated fat accumulation in later adulthood (Fig 1D, bottom left). When expressed in percentage, the composition of lean mass shows a steady increase up to three months of age in $Pw1^{+/+}$ males (Fig 1D, top right), while fat mass decrease comparatively (Fig 1D, bottom right). These male-specific changes in body composition were less prominent in $Pw1^{+/pat-}$ males. Furthermore, the phenotype is highly specific to male, as lean and fat mass were proportionally reduced in the $Pw1^{+/pat-}$ females as compared to their $Pw1^{+/+}$ littermates. Therefore, there was no difference in % of body composition in females (Fig 1D, right). Two-way ANOVA test revealed an interaction between genotype and

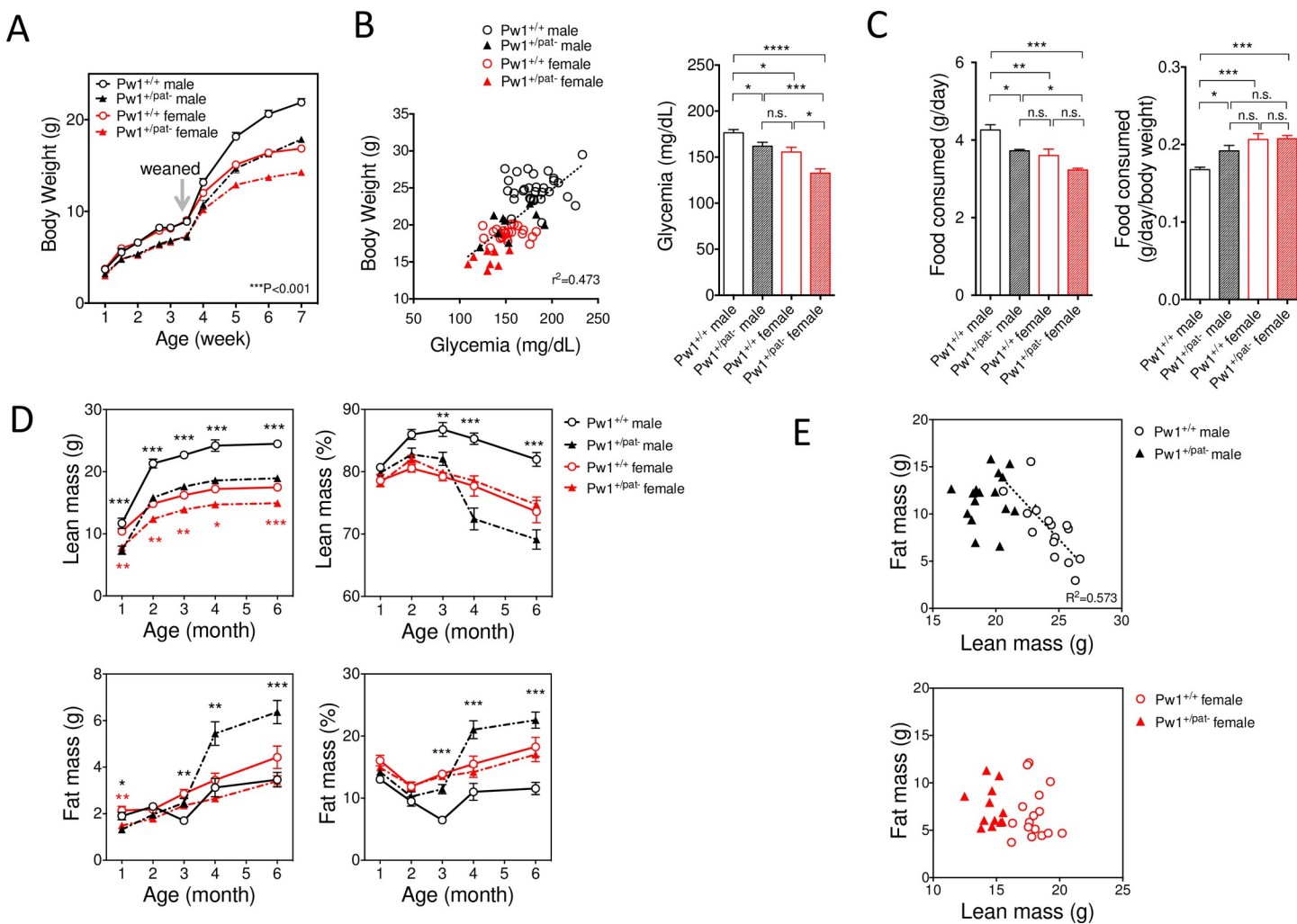

**Fig 1. Reduced masculinization of metabolisms in *Pw1^+/pat-* males.** (**A**) Postnatal growth of *Pw1^+/pat-* compared with *Pw1^+/+* littermates showing Pw1^+/pat- animals are smaller in both sexes throughout the growing phase (p<0.001), and the onset of sexual dimorphism in body weight is delayed in Pw1^+/pat- at four weeks of age. Two-way ANOVA test revealed significant interaction between genotype and sex after 6 weeks of age. (**B**) Random-fed glucose levels at 2 month-old and its positive correlation with body weight (r = 0.690, p<0.0001). Each symbol represents independent measurement. (**C**) Sex dimorphic food consumption at 2 month-old. Two way ANOVA test with multiple comparisons demonstrated significant interaction between sex and genotype. (**D**) Sexual dimorphisms in body composition in young adults (n = 8–14 each group). Male-specific increase in lean mass and decrease in fat mass in *Pw1^+/+* males at 2–3 month of age was less prominent in *Pw1^+/pat-* males, while the females *Pw1^+/pat-* are proportionally smaller in lean and fat mass. * in black represents comparison in males and * in red represents comparison in females. (**E**) An inverse correlation between lean mass and fat mass at 10 month of age was only found in the *Pw1^+/+* males (r = 0.757, p<0.001) and not in the *Pw1^+/p-* males (r = 0.076, p = 0.771) nor in the females (r = 0.006, p = 0.981). Comparison between genotypes and sexes was performed using two-way ANOVA with Tukey's multiple comparisons. Correlation was determined with simple linear regression analysis. *P<0.05, **P<0.01, and ***P<0.001. NS: non-significant. Values are mean ± SEM. Each symbol represents individual animals in (**B**) and (**E**).

sex with age, indicating that loss of *Pw1* has significantly different impacts on body composition in males and females. There was an inverse correlation between lean mass and fat mass in mature age specifically in *Pw1^+/+* adult males (r = 0.757, p<0.001) (Fig 1E). Taken together, *Pw1^+/pat-* mice displayed a significant reduction in male-specific body growth.

## *Pw1* deficient mice have altered GH/IGF signaling that reduces body size and sexual dimorphism during postnatal development

The growth hormone (GH)/insulin-like growth factor-1 (IGF-1) axis plays a pivotal role in directing postnatal growth and regulates fat metabolism [33], whereas gonadal androgens

stimulate the male-specific pulsatile secretion of GH in early puberty [34,35] thereby promoting sexually dimorphic patterns of somatic growth and body composition. The anabolic effect of GH is exerted by the stimulation of endocrine IGF-1 production primarily in the liver, and the circulating IGF-1 levels are considered as an indicator of GH activity in the postnatal growth phase [33]. Therefore, we examined IGF-1 activity in the Pw1 mutant mice during postnatal development. Plasma IGF-1 levels correlated with body weight at 3 weeks old, as commonly expected (Fig 2A), and IGF-1 levels were reduced in the Pw1$^{+/pat-}$ mice as compared to Pw1$^{+/+}$ littermates (Fig 2B). We further monitored the circulating levels of IGF-1 in the same animals weekly up to 6 weeks of age corresponding to the period when the circulating IGF-1 levels dynamically change in a sex-dependent manner [36]. At five weeks of age, IGF-1 levels were significantly different between sexes (p<0.001) when IGF-1 levels decline in females corresponding to an earlier cessation of growth and increase in males to further promote their growth. Therefore, the levels of IGF-1 in Pw1$^{+/+}$ males were significantly higher as compared to Pw1$^{+/+}$ and Pw1$^{+/pat-}$ females (p<0.05 and p<0.01, respectively). Remarkably, no statistical differences were found between Pw1$^{+/pat-}$ males and females of both genotypes. We conclude that the Pw1$^{+/pat-}$ animals display reduced sex-specific regulation in IGF-1 secretion compared to the wild-type littermates.

Based on the observation that circulating IGF-1 levels are reduced in Pw1$^{+/pat-}$ mice at 3 weeks of age, we performed gene expression analysis on pituitary gland and liver of different sets of littermates. Consistently, the expression of Gh in the pituitary gland was significantly reduced in Pw1$^{+/pat-}$ males as compared to Pw1$^{+/+}$ littermates, whereas growth hormone receptor (Ghr) and Igf1 expression levels in the liver also showed a trend of down-regulation (Fig 2C). Taken together, these results show a global suppression of GH/IGF-1 activity during postnatal growth in Pw1$^{+/pat-}$ mice in a sex-dependent manner.

## Pw1 deficiency deregulates insulin sensitivity and increases adiposity in adult males

Insulin is a key regulator of energy and fat metabolism throughout life. Its anabolic action promotes postnatal growth after weaning [37], however, chronically elevated insulin levels are associated with obesity and abnormal fat metabolism [38]. To evaluate the steady state insulin levels, we measured blood insulin levels in fed animals. At 3 months of age, circulating insulin levels were lower in females than males (p<0.001) corresponding to their lower levels of glycemia (Fig 2D). The insulin levels of Pw1$^{+/pat-}$ were also reduced at this age although this trend is only confirmed with Fisher's LSD test. By contrast, the Pw1$^{+/pat-}$ genotype exhibited a male specific increase in insulin levels at 6 months of age.

Linear regression between insulin levels and body composition revealed a positive correlation between plasma insulin levels and lean mass at 3 months of age in Pw1$^{+/+}$ and Pw1$^{+/pat-}$ males (r = 0.714, p<0.01, and r = 0.699, p<0.01), respectively (S1A Fig). At 6 months of age, on the other hand, the insulin levels correlated better with fat mass in Pw1$^{+/+}$ and Pw1$^{+/pat-}$ males (r = 0.875, **p<0,01 and r = 0.644, p = 0.118, respectively).

An insulin tolerance test (ITT) was performed in a set of Pw1$^{+/+}$ and Pw1$^{+/pat-}$ male littermates, which revealed no differences between genotypes at 3 months of age, whereas the Pw1$^{+/pat-}$ males developed a modest insulin resistance at 6 months of age as compared to Pw1$^{+/+}$ males (Fig 2E). Oral glucose tolerance test (OGTT) on the animals of the same litter showed that insulin secretion and glucose clearance were slightly lower in Pw1$^{+/pat-}$ at 3 months of age (Fig 2F). Notably, these patterns were inverted at 6 months of age and Pw1$^{+/pat-}$ males displayed a higher insulin secretion and clearance.

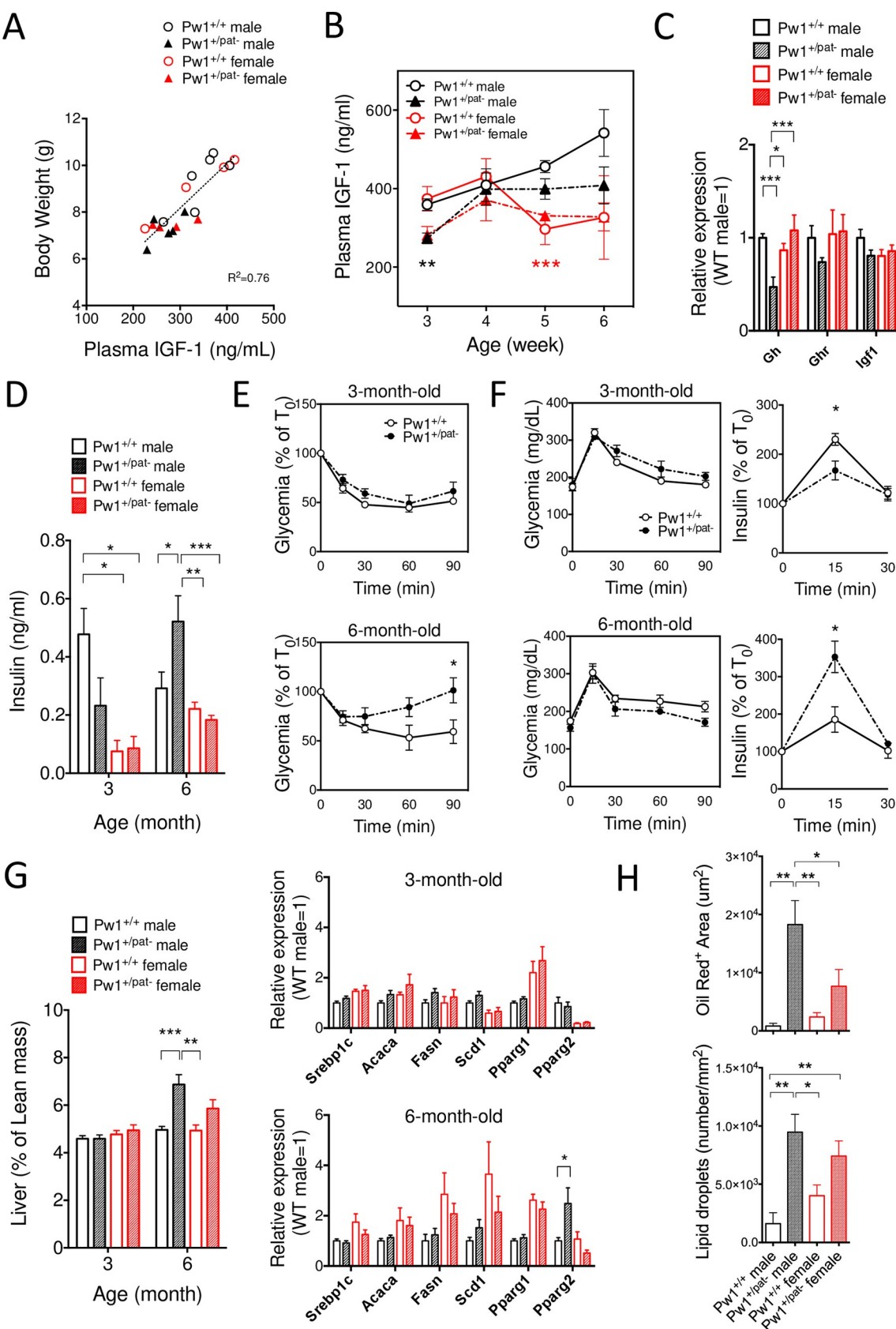

**Fig 2. Deregulated GH/IGF axis in *Pw1*$^{+/pat-}$ youngs and insulin homeostasis in *Pw1*$^{+/pat-}$ adult males.** (**A**) Circulating IGF-1 levels at 3 weeks of age and its positive correlation with body weight. Correlation was determined with simple linear regression analysis. (**B**) Blunted sexual dimorphism of circulating IGF-1 dynamics in *Pw1*$^{+/pat-}$ mice as compared to that of *Pw1*$^{+/+}$. N = 4–8 each group from 5 litters. Two-way ANOVA showed that the IGF-1 levels at 3 weeks old are significantly lower in Pw1$^{+/pat-}$ mice, with no difference between sexes. **P<0.01 in black represent comparison between genotypes, whereas ***P<0.001 in red represent comparison between sex. (**C**) mRNA expression of growth hormone (*Gh*) in the pituitary gland, and GH receptor (*Ghr*) and insulin-like growth factor (*Igf1*) in the liver at 3 weeks of age. Values were normalized with Tbp and presented relative to the *Pw1*$^{+/+}$ male littermates (n = 6–12 each group). (**D**) Random-fed blood insulin levels at 3 months and 6 months of age in *Pw1*$^{+/pat-}$ and *Pw1*$^{+/+}$ littermates (n = 4–6). (**E**) Representative insulin tolerance test (ITT) on *Pw1*$^{+/pat-}$ and *Pw1*$^{+/+}$ males from a single litter (n = 3 each genotype) at 3 months and 6 months of age. Similar results were obtained from two other litters. (**F**) Representative oral glucose tolerance test (OGTT) with insulin secretion measurement on the same set of mice as in ITT. (**G**) Liver size and mRNA expression of lipogenic genes in the liver at 3 months and 6 months of age showing an age-dependent development of hepatic steatosis in *Pw1*$^{+/pat-}$ mice as compared to *Pw1*$^{+/+}$ littermates. Srebp1, sterol regulatory element binding protein 1; Acaca, acetyl-CoA carboxylase alpha; Fasn, fatty acid synthese; Scd1, Stearoyl-CoA desaturase; Pparg1 & 2, peroxisome proliferator activated receptor gamma 1 & 2. (**H**) Fat deposition revealed by Oil Red-O staining in 8-month-old livers (n = 4–6 for each group). Lipid droplets were quantified in number and in size using particle analysis tool in Image-J software. Values are mean ± SEM; *, P < 0.05; **, P < 0.01; ***, P < 0.001 by two-way ANOVA with Tukey's multiple comparisons.

*Pw1* reporter gene expression was high in pancreatic β-cells and in hepatocytes (S2A and S2B Fig), both of which were characterized by the presence of sex steroid hormone receptors [39,40]. Co-localization of *Pw1* with ERα in various endocrine cells indicates a pivotal role of *Pw1* in these cell types via sex steroid signaling.

Paternal loss of *Pw1* has been shown to lead to increased β-cells cycling in *Pw1*$^{+/pat-}$ males at 3 months of age [41]. The increase of proliferation at a younger age may result in increased insulin production in later adulthood. We analyzed the insulin content of pancreas in mature adult males and observed that the pancreatic insulin is slightly elevated in the *Pw1*$^{+/pat-}$ males (S1B Fig, top). In addition, random-fed glycemia was significantly elevated in *Pw1*$^{+/pat-}$ males, in agreement with their insulin resistance in adulthood (S1B Fig, bottom).

*Pw1*$^{+/pat-}$ animals also exhibited age- and sex-dependent hepatic phenotypes: liver size was significantly higher in *Pw1*$^{+/pat-}$ males at 6 months of age as compared to *Pw1*$^{+/+}$ males and females (Fig 2G, left). Gene expression of major adipogenic genes in these animals demonstrated significant changes in the 6-month-old *Pw1*$^{+/pat-}$ livers in a sex-dependent manner (Fig 2G, right). Notably, the two major isoforms of *Pparg1* and *Pparg2*, differentially expressed between males and females [42], were differently affected by *Pw1* loss of function. While the *Pparg1* is similarly expressed in *Pw1*$^{+/+}$ and *Pw1*$^{+/pat-}$ livers, *Pparg2* expression levels were significantly increased in 6-month-old male livers. In contrast, *Pparg2* levels were significantly lower in female livers as compared to male livers at 3 months of age and no increase was observed in *Pw1*$^{+/pat-}$ female livers at 6 months of age.

*PPARG2* is selectively increased in human obesity [43] and is specifically elevated in the steatotic livers of *ob/ob* mice [44]. We therefore performed hepatic histology using Oil Red-O staining on the 8-month-old livers of both sexes (Figs 2H and S1C). *Pw1*$^{+/+}$ livers revealed multiple small lipid droplets in both sexes. In contrast, *Pw1*$^{+/pat-}$ mice showed abundant, large lipid droplets that were more marked in males. Digital quantification revealed that the total number of lipid droplets and Oil Red-O positive area size were significantly increased in *Pw1*$^{+/pat-}$ livers (p<0.001) in mature adulthood (Fig 2H). We note that smaller droplets are more abundant in *Pw1*$^{+/+}$, whereas larger droplets increased by age in *Pw1*$^{+/pat-}$ livers, and that this trend was more pronounced in males (S1C Fig).

Taken together, our findings demonstrated that paternal *Pw1* deficiency affected multiple stages of early life that proceeded to an age- and sex-dependent global shift of body metabolism towards accelerated adiposity, diabetic-like insulin resistance, and fatty liver in later adulthood, and the impact is more profound in males.

## Paternal *Pw1* deficiency reduces aggressive behavior and social dominance in males

During routine handling of the *Pw1* mutant colony, we observed that adult *Pw1*$^{+/pat-}$ males seldom display typical aggressive behavior as compared to their *Pw1*$^{+/+}$ littermates. We scored incidents of spontaneous fights among *Pw1*$^{+/+}$ (n = 75) and *Pw1*$^{+/pat-}$ (n = 57) male offspring that were group-caged with littermates, and found that *Pw1*$^{+/pat-}$ males were significantly less aggressive (S3A Fig). When male offspring were separated according to genotype at the time of weaning (*Pw1*$^{+/+}$ or *Pw1*$^{+/pat-}$), we observed little incidents of fight in the *Pw1*$^{+/pat-}$ cages, suggesting that the reduced aggressive behavior is, at least in part, intrinsic to the paternal *Pw1* loss of function.

To quantitatively assess the competitive ability of *Pw1*$^{+/pat-}$ males, we used a social-confrontation tube test [45] on adult offspring derived from *Pw1*$^{+/pat-}$ breeder males. The first test was to examine whether *Pw1* is involved in establishing social hierarchy among littermates by using litters consisting of two genotypes. We observed a typical social dominance pattern in which *Pw1*$^{+/+}$ males dominate the siblings in a given cage at 10 months of age (S3B Fig, squared in red). Each animal was ranked within each litter by the number of wins and the score was compared between genotypes. This ranking revealed that the *Pw1*$^{+/+}$ males rank higher and there is a significant difference between genotype (***p<0.001) (Fig 3A). Notably, the same analysis on younger litters at 3 months of age revealed no significant difference in the inter-litter rank between genotypes, suggesting that younger males have not yet established social rank at this age. A second test was performed in the context of stranger encounter as described by Garfield et. al. [46] in which animals were tested against unfamiliar opponents from different cages with mixed genotypes. The winning rate was determined by the percentage of win in all matches against unknown opponents (S3B Fig). This test demonstrated that the *Pw1*$^{+/+}$ males have a greater likelihood of winning in a forced encounter (*P<0.05) (Fig 3B).

In the female *Pw1*$^{+/p-}$ brains, the oxytocinergic architecture appeared under-developed concomitant with alteration in maternal care [28], a female specific behavior that is acquired at pregnancy. On the other hand, the aggressive behavior commonly observed in laboratory mice is male specific and develop during postnatal growth period. Oxytocin and arginine-vasopressin (AVP) are the two major neuropeptide that regulates sex-specific mammalian behaviors (reviewed in [47,48]). In particular, the AVP system is androgen-dependent [49] and central AVP plays a pivotal role in inter-male aggressive behavior [50,51]. *Pw1* is shown to be expressed in both oxytocinergic and vasopressinergic neurons [52]. Therefore, we hypothesized that *Pw1* plays a pivotal role in regulating the function of these cell types through sex hormone signaling. We first examined *Pw1* expression in the brain using the *Pw1* reporter transgenic mouse line Pw1$^{IRESnLacZ}$ [17]. As predicted, we found high levels of reporter gene expression in brain nuclei known to be sexually dimorphic and express sex steroid hormone receptors [53,54], including paraventricular nucleus (PVN) of hypothalamus, the bed nucleus of stria terminalis (BnST), the medial preoptic area (mPOA), and the medial amygdala (MeA) (S3C Fig). These regions are primary sites of AVP production and vasopressinergic neuronal projections [50,55]. We therefore examined the brains from *Pw1* reporter mice by immunofluorescence using anti-β-gal and anti-AVP antibodies and found that the vasopressinergic cells in the PVN and SON are the sites of high *Pw1* reporter gene expression (Fig 3C, top) which co-express ERα (Fig 3C, bottom), suggesting a role of *Pw1* in this cell type. We next examined the architecture of AVP$^{+}$ cells in the *Pw1*$^{+/+}$ and *Pw1*$^{+/pat-}$ males whose competitive ability had been already established by the tube test (Litter 1–6 in S3B Fig). Using anti-AVP antibody, we immunostained the coronal sections of entire brain and the total AVP$^{+}$ cell number in the

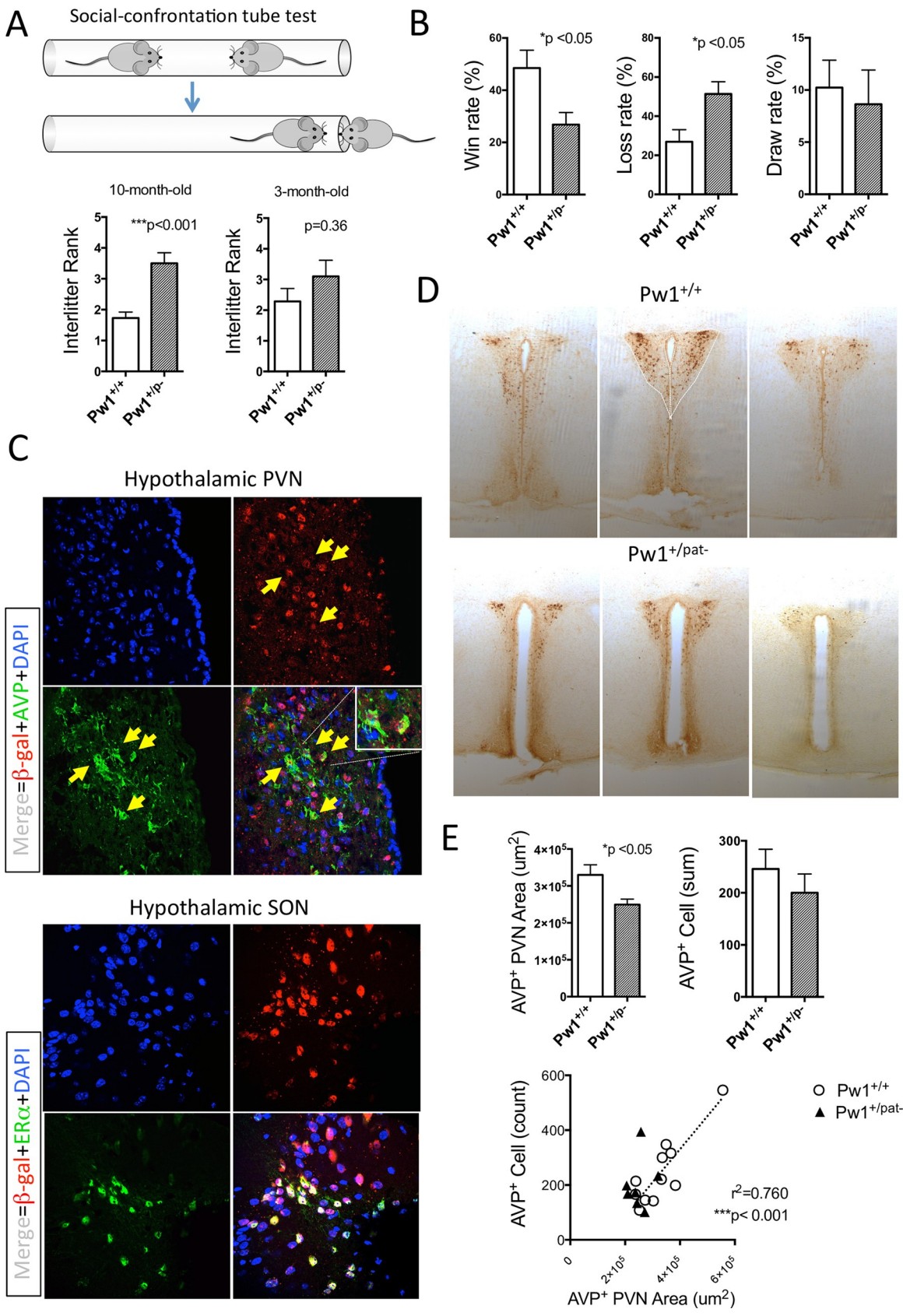

**Fig 3. Altered social behavior and brain architecture in *Pw1*$^{+/pat-}$ males.** (**A**) Interlitter social rank by tube test in *Pw1*$^{+/+}$ and *Pw1*$^{+/pat-}$ males from mixed genotypes at 10 months of age (n = 11 vs n = 10, from 5 litters) and at 3 months of age (n = 7 vs n = 10, from 4 litters). (**B**) Assessment of social dominance in the stranger encounter tube test. Animals used were listed in S3B Fig. The winning rate was calculated from 17–18 matches against unfamiliar opponents. (**C**) *Pw1* reporter expression (β-gal$^+$) is observed in the vasopressinergic neurons of PVN (top), whereas β-gal signals are strongly co-localizing with ERα receptor in SON (bottom) in the hypothalamus (x400). (**D**) Representative images of AVP expressing neurons in the PVN of hypothamamus in the *Pw1*$^{+/+}$ and *Pw1*$^{+/pat-}$ male brains (x40). Coronal sections at 120μm intervals through PVN from anterior to posterior axis were immunostained with an anti-AVP antibody. AVP positive area size (dotted line) and cell count were quantified from five sequential sections. E. Digital quantification of AVP$^+$ area size and the total cell count in PVN and their positive correlation in the Pw1$^{+/+}$ male brain. Columns, mean; bars, SEM; *, P < 0.05, Mann-Whitney U test.

PVN and its area size were determined (Fig 3D). Concordant with the reduced social competitiveness, the AVP$^+$ PVN area size was significantly reduced in the *Pw1*$^{+/+}$ brains at 10 months of age (Fig 3E). In the *Pw1*$^{+/+}$ brain, we found a strong positive correlation between the area size and the cell number (Fig 3E). Remarkably, this correlation is abolished in the *Pw1*$^{+/pat-}$ brain, implying that the proliferation and/or expansion of the AVP$^+$ cells are deregulated. Finally, we examined the correlation between the AVP$^+$ cell structure and social behavior in the litter 1 which consists of four *Pw1*$^{+/+}$ males. Both area size and cell count in the PVN showed positive correlation with the winning rate in this set of animals (S3D Fig).

These data suggests that *Pw1* promotes acquired social dominance and aggressive behavior by modulating the AVP$^+$ neuroendocrine architecture in male mice.

## *Pw1* promotes testosterone production in young male mice

Perinatal androgen secretion leads to changes in the CNS and underlie sexual dimorphism in the brain [56]. The testis produce androgens in adolescence that contribute to the development of adult male characteristics in metabolism and behavior that typically underlies reproductive success including male mating and aggression [57–59]. We therefore measured levels of testosterone in young males during peripubertal development when testosterone secretion peaks in postnatal growth (between 6–9 weeks)[60]. First, we measured testosterone levels of male *Pw1*$^{+/+}$ and *Pw1*$^{+/pat-}$ littermates derived from 3 litters and found that the *Pw1*$^{+/pat-}$ mice display a delayed pubertal surge (Fig 4A). Based on this timing, we further analyzed different young males from multiple cages containing mixed genotypes in the entire colony at the age between 2.5 and 3 month old. These analyses revealed that *Pw1*$^{+/pat-}$ males have significantly lower levels of testosterone as compared to *Pw1*$^{+/+}$ males (Fig 4B). In agreement with these observations, mRNA expression of steroidogenic genes encoding the rate limiting enzymes for testosterone biosynthesis (Cyp17a, Cyp11a, and 3β-HSD), as well as that of luteinizing hormone receptor (LHR) in the testes are significantly reduced in *Pw1*$^{+/pat-}$ males (Fig 4C). Notably, there was no significant difference in the expression of aromatase Cyp19 or 17β-HSD3.

A previous study from our laboratory using the *Pw1* reporter transgenic mouse line (Pw1$^{IR-ESnLacZ}$) demonstrated that *Pw1* is highly expressed in the peritubular cells near the basement membrane of seminiferous tubules, a part of which were identified as Bmi1$^+$ spermatogonia [17]. Since the *Pw1*$^{+/pat-}$ mice displayed reduced steroidogenesis, we were interested in whether *Pw1* is expressed in the cell types responsible for the testosterone production in the testis. We performed histological analyses on the testis of Pw1$^{IRESnLacZ}$ mice using an anti-β-gal antibody and antibodies against sex steroid hormone receptors and found that reporter gene expression colocalized with androgen receptor expression (Fig 4D). In the testis, Leidig cells and Sertoli cells are the two cell types predominantly express androgen receptor (AR) [61]. Therefore, we conclude that *Pw1* reporter gene is abundantly expressed in the

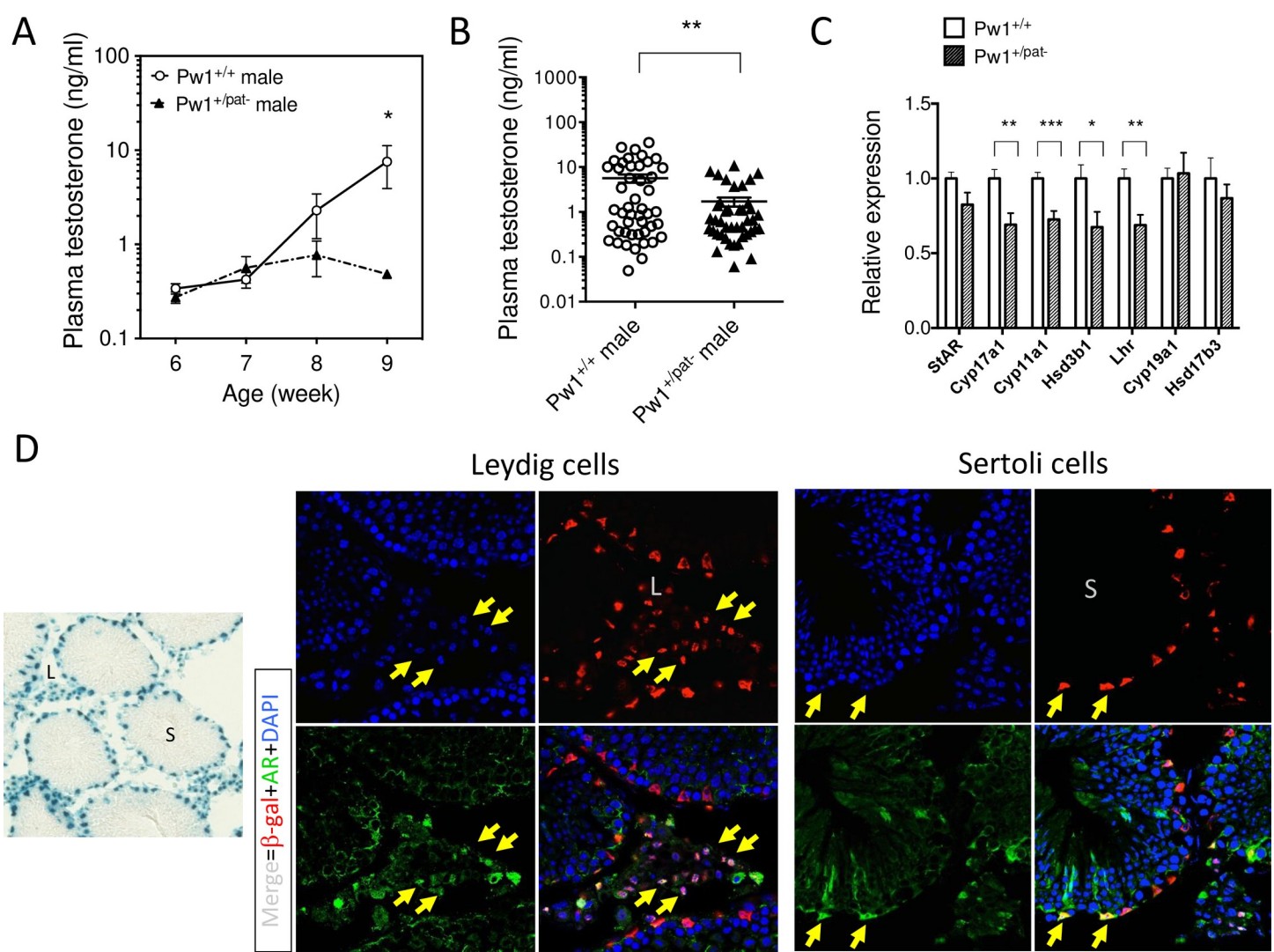

**Fig 4. Reduced testosterone production in the *Pw1^+/pat-* males.** (**A**) Plasma testosterone measurements at peripubertal age in *Pw1^+/+* and *Pw1^+/pat-* males (n = 7 and 5, respectively, from 3 litters). Two-way repeated measures ANOVA with Sidak's multiple comparisons test revealed a significant difference between genotypes at 9 weeks of age (*P<0.05). (**B**) Plasma testosterone levels in *Pw1^+/+* and *Pw1^+/pat-* non-breeder littermates (n = 47 and n = 42) at 2.5–3 months of age. Each symbol represents independent animals. Bars represent mean ± SEM; **, P<0.01, by Mann Whitney U test. (**C**) mRNA expression of steroidogenic genes in the 2.5–3 month-old testes of *Pw1^+/+* and *Pw1^+/pat-* non-breeder littermates (n = 15 versus n = 11). StAR, Steroidogenic acute regulatory protein; Cyp17a1, cytochrome P-450 17a; Cyp11a1, cholesterol side-chain cleavage enzyme; Hsd3b1, 3-b-hydroxysteroid dehydrogenase; Lhr, luteinizing hormone receptor; Cyp19a1, aromatase enzyme; Hsd17b3, 17-b-hydroxysteroid dehydrogenase. (**D**) *Pw1* reporter expression in the 2.5-month-old testis of Pw1^IRESnLacZ mice, showing X-gal staining (x100) and β-gal immunofluorescence (x400) in the interstitial compartment and in the epithelium of seminiferous tubules. The β-gal immunofluorescence overlapped with that of androgen receptor (AR) in the Leydig and Sertoli cells (arrows). L, interstitial Leydig cells; S, seminiferous tubule.

testosterone producing Leydig cells in the interstitial compartment and in the supporting Sertoli cells near the basement membrane of seminiferous tubules. We note that other imprinted genes are highly expressed in the Leidig cells and that a number of imprinted genes are simultaneously deregulated in Leidig cells of human patients with idiopathic germ cell aplasia [62].

Taken together, our data demonstrate that male sex-hormone signaling during secondary sexual maturation is suppressed in *Pw1^+/pat-* male mice, concomitant with reduced gene expression for testosterone biosynthesis in the testis, which may account for the reduced masculinization in metabolism and social behavior during the postnatal development.

## Discussion

We demonstrate that *Pw1* plays a key role at specific stages of postnatal development for sex-specific growth, metabolism and behavior. *Pw1* deficient male mice exhibit a smaller body size and a reduced masculinization of body composition followed by a global shift of metabolism leading to early onset obesity and related metabolic changes. A marked reduction of growth promoting GH/IGF-1 and insulin characterized early postnatal life and a decrease in testosterone activity at puberty coupled with insulin-resistance lead to a male-specific deficiency by adulthood (Fig 5). These observations are concomitant with a significantly reduced sexual dimorphism in body composition. In addition, our data suggests a model in which a paternally expressed gene promotes male-specific brain development and behavior through a sex steroid pathway that contribute to reproductive success in mammals. Our finding support the intralocus sexual conflict model of genomic imprinting [6] for the control of mammalian sexual dimorphism.

The physiopathological alterations in *Pw1* deficient mice identified in this study are similar to the phenotypic spectrum of Prader-Willi syndrome (PWS), a disorder with hypothalamic dysfunction and hypopituitarism due to imprinting errors [63]. PWS is characterized by biphasic clinical manifestations, i.e., reduced growth velocity, hypoglycemia, and hypotonia at infancy, followed by hyperphagia, extreme obesity, and hypogonadism in childhood through adulthood. In addition, patients present low levels of GH and IGF-1, gonadotropins, and gonadal sex steroids, hence, the established treatment includes GH administration at infancy and sex hormone replacement in young adult that improve their growth and metabolism by enhancing muscle mass and reducing fat mass [64].

We found that the expression and function of *Pw1* are particularly concentrated in the hypothalamus, pituitary gland, and gonads. Accordingly, our findings suggest that a primary *Pw1* deficiency involves hypothalamic-pituitary-gonadal dysfunction at critical periods of postnatal development, leading to changes in sex steroids levels and GH/IGF signaling. Our observations are in line with several studies reporting that mouse mutants for paternally expressed genes typically display hypothalamo-pituitary phenotypes [65,66]. For instance, a loss of *Grf1* or *Dlk1* resulted in reduced GH content and secretion [67,68], while disruption of the imprinting domain encompassing *Dlk1*, *Rtl1*, and *Dio3* resulted in transient perturbation in the GH/IGF-1 growth pathway with hypothyroidism [69]. Together, these observations pointed that imprinted genes play pivotal roles for the regulation of hypothalamo-pituitary-gonadal (HPG) axis governing growth, metabolism and reproduction.

Sexual dimorphism in mammalian growth is regulated by the sex steroid-dependent GH/IGF-1 pathway. During postnatal development, testosterone and 17-estradiol evoke pituitary production of GH in discrete manners [70,71]. GH in turn stimulates systemic IGF-1 through the JAK2/Stat5b pathways [72,73]. Estrogen-bound estrogen receptor (ER) also induces transcription of IGF-1 and IGF receptor through estrogen-responsive elements at their gene loci [74]. The activation of IGF-1/IGF1R signaling further activates transcription factors including ERs, forming a complex crosstalk between IGF-1 and sex steroid signaling that amplifies somatic growth in puberty [75]. Notably, sex differences during postnatal growth are diminished in the *Stat5b* deficient mice [76], the GH receptor (*Ghr*)- and *Igf1*-deficient mice [77]. Hence, reduced sexual dimorphism in *Pw1* deficient young is likely due to an attenuated hypothalamic regulation of sex hormones and downstream GH/IGF-1 signaling.

Androgens and their cognate receptors regulate male secondary sexual differentiation [78], and testosterone deficiency is related to metabolic syndromes in men [79,80]. In androgen receptor deficient (ARKO) mice, late-onset obesity and fatty liver, as well as insulin- and leptin-resistance have been reported which is restricted to males [81–83]. Elevated levels of

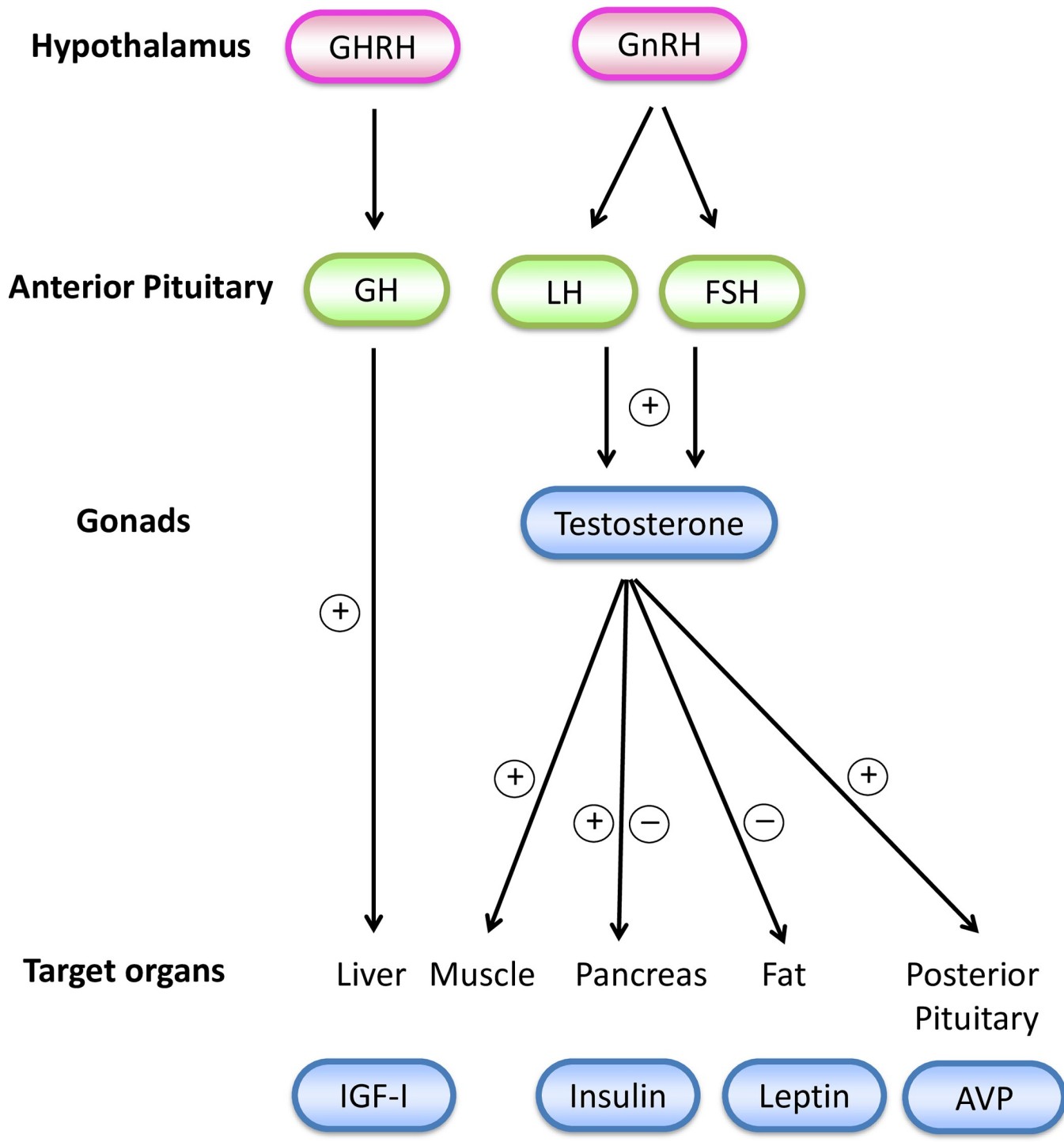

**Fig 5. Hormonal cascades are deregulated in *Pw1*$^{+/pat-}$ mice.** *Pw1* is abundantly expressed in the gonads and target organs and colocalizes with steroid hormone receptors in hormone secreting cells. GHRH, growth hormone-releasing hormone; GH, growth hormone; IGF-1, insulin-like growth factor 1; GnRH, gonadotropin-releasing hormone; LH, luteinizing hormone; FSH, follicle-stimulating hormone; AVP, arginine vasopressin; +, stimulation; -, inhibition by *Pw1*.

circulating leptin and leptin-resistance were also reported previously in the $Pw1^{+/pat-}$ adult males [31]. Leptin was shown to reciprocally regulate gonadal hormone production at central and peripheral levels [84] and the steroidogenesis in the Leydig cells was impaired in the leptin-deficient *ob/ob* males [85] concomitant with loss of skeletal masculinization [86]. Therefore, the defect in leptin signaling in $Pw1^{+/pat-}$ males may causally relate to further reduction in gonadal steroidogenesis. We note, however, that the *Pw1* deficient phenotypes were not restricted to males, as both $Pw1^{+/pat-}$ males and females displayed delayed postnatal growth until weaning. Smaller body size of $Pw1^{+/pat-}$ mice before weaning may originate, at least in part, from changes arising during prenatal development, as observed by the lower body weight and myofiber numbers at birth [19]. Sex specificity of resulting phenotypes upon *Pw1* loss is presumably due to sex dimorphic postnatal dependency to sex steroid signaling in males and females. Sex-dependent phenotypes in *Pw1* deficiency have also been demonstrated in placental function, where female placenta compensates the loss of *Pw1* better than male placenta [30,87]. This indicates that the sex-dependent *Pw1* function starts much earlier than the offspring's postnatal manifestation of sexually dimorphism in growth.

Testosterone exposure at specific phases of development, such as perinatal and pubertal periods, is crucial for the brain masculinization [56,58]. This study shows that *Pw1* expression and sex steroid receptors highly overlap in key endocrine and neural tissues that regulate sex-specific behaviors. Specifically, we found that paternal loss of *Pw1* function attenuated social dominance and correlates with the decreased expansion of the testosterone-sensitive vasopressinergic nuclei in the hypothalamus. In general, the sexually dimorphic nuclear volume/neuron number is ascribed to the sex steroid control of proliferation and cell death [88]. Given that *Pw1* regulates cell survival and/or apoptosis at a cellular level, it is conceivable that *Pw1* participates in sex-specific brain formation and neuronal proliferation/differentiation for various acquired behaviors. In contrast, aggression in males is ERα dependent [89] and the intermale competitive ability reflects the level of ERα expression in BnST [45]. *Pw1* may contribute to masculinization by regulating the expression of cognate receptors in sex-hormone dependent cells, thereby determining the dependency of organisms on sex steroid signaling.

Our finding suggests that *Pw1* plays a role in regulating the proliferative and/or cell number in various tissues through sex steroid signaling during tissue remodeling in adult life. We demonstrate further that *Pw1* expressing adult stem/progenitor cells are characterized by the presence of sex steroid receptors. Consistent with this observation, the number of *Pw1* expressing cells in the heart were shown to increase during pregnancy [90], while the pregnancy-induced expansion of pancreatic β-cells inversely correlates with the levels of *Pw1* expression [41]. Accumulating evidence also demonstrates the interaction of sex hormones in stem cell behaviors. Kim *et. al.* demonstrated that sex steroid hormones promote the establishment of the adult quiescent satellite cell pool through the HPG axis [91]. Likewise, ERα signaling has been shown to drive pancreatic β-cell replication in development as well as in post-injury regeneration [92]. How *Pw1* loss of function alters adult stem/progenitor cell behaviors or steroid hormones *per se* affects pluripotency is beyond the scope of this study, however, we have shown previously that loss of *Pw1* function leads to a loss in self-renewal capacity in adult muscle stem cells [19] as well as skin stem cell [17]. The link between self-renewal, stem cell competence and sex steroid hormones presents an emerging concept in stem cell biology.

This study uncovers a key role of the parentally imprinted gene *Pw1* in establishing male-specific characteristics in metabolism, brain structure and behavior by modulating sex steroid pathways. We note that similar metabolic phenotypes observed in *Pw1* deficient mice were reported in the progeny of human populations that have been subjected to overnutrition or prolonged nutritional deprivation [93], coinciding with a deregulation of several parentally imprinted genes [94]. Our findings show that *Pw1* provides an epigenetic context that links

energy metabolism to the generation of male-specific traits in mice. Analogous mechanisms may also underlie the sex differences in the metabolic syndrome in humans.

## Materials and methods

### Ethics statement

All experiments were in adherence to the institutional guideline for experiment and husbandry of laboratory animals. Approval for the animal (mouse) work performed in this study was obtained through review by the French Ministry of Education, Agreement #A751320.

### Animals

The Pw1$^{IRESnLacZ}$ reporter mice [17] and the *Pw1* deficient mice carrying targeted mutation Pw1Δ9 [29] were generated in our laboratory and maintained in C57BL/6J genetic background. Eight *Pw1$^{+/pat-}$* males (two generation after the first deletion of *Pw1* allele, designated F3) were used as breeders. In order to compare *Pw1$^{+/pat-}$* offspring to their *Pw1$^{+/+}$* littermates, *Pw1$^{+/pat-}$* breeder males under the age of 8 months old were crossed with young C57BL/6J females (purchased from Janvier Laboratory, Le Genest St Isle, France), and large litters with both genotypes were pooled for each analysis whenever available. All metabolic phenotypes presented in this study were obtained from offspring of F3-F4, and behavior phenotypes from F5 generations. Offspring were weaned at the age between 3–4 weeks, and housed in groups with littermates unless otherwise stated (maximum 6 per cage). Mice were kept on 12-hour light/dark cycle at 24˚C with *ad libitum* access to water and standard chow diet.

### Postnatal growth and body composition

Postnatal growth of *Pw1$^{+/pat-}$* mice was measured along with *Pw1$^{+/+}$* littermates, which were nursed together by C57BL/6J females. Longitudinal body composition analysis was realized using Brucker minispec nuclear magnetic resonance (NMR) imaging (Brucker, USA), starting from 1 month of age. The lean and fat mass expressed in percentage (%) was converted to weight (g) in respect to the total body mass (g).

### Blood glucose and food intake

Blood was sampled from the tail tip and glycemia was measured using an Accu-Check glucometer with disposable test strips (Roche Diagnostics, Basel, Switzerland). Food intake was assessed on N = 5–7 mice per genotype per sex, individually housed at 8 weeks of age and food consumption was measured twice a week for 3 weeks.

### Plasma hormone measurement

Blood was collected either from the tail tip or from the facial vein of randomly fed animals at dormant period of circadian rhythms (11am-16pm), using Microvette CB300 capillary action blood collection tubes (Sarstedt, Nümbrecht, Germany). Plasma IGF, insulin, and testosterone levels were determined using ELISA kits for mouse (Chrystal Chem. Ltd, NJ, USA).

### Glucose and insulin tolerance test

Oral glucose tolerance test (OGTT) and insulin tolerance test (ITT) were performed on 5–6 hours fasted mice. D-glucose (2 g/kg) was orally administered, and human insulin (0.25–0.55 units/kg) was intraperitoneally injected. For both tests, tail blood was sampled at 0, 15, 30, 60, and 90 minutes after the administration, and glycemia was immediately measured as described

above. For monitoring insulin secretion upon OGTT, tail blood was collected at 0, 15, 30 minutes and subjected for ELISA protocol.

## Quantification of fatty liver

To visualize lipid droplets, Oil red O-staining was performed in the 10um frozen sections of 8-month-old liver. Number of lipid droplets and the total oil Red positive area were quantified using 'particle measurement' in the Image-J software [95].

## Gene expression analysis

Total RNA was isolated using TRIzol reagent (Ambion) and subjected to cDNA synthesis (Lifetechnology), followed by quantitative PCR analysis (LightCycler, Roche). The expression levels of target genes were normalized with the levels of a housekeeping gene coding TATA binding protein (*Tbp*). Primer sequences are available upon request.

## Assessment of social dominance by confrontation tube test

We applied the social confrontation tube test previously described by Garfield *et. al.* on *Grb10* deficient mice [46] to determine social dominance among $Pw1^{+/+}$ and $Pw1^{+/p-}$ males. All mice used in the test were group-housed with littermates consist of both $Pw1^{+/+}$ and $Pw1^{+/p-}$ genotypes (n = 4–6) at mature age between 10–12 months. Independent sets of mice were analyzed at 3 months old for comparison.

   To determine the social hierarchy within a given litter, animals from a given litter were challenged with their littermate. On subsequent days, the same animals were also challenged with unfamiliar opponents from different litters. On the day of experiment, the animals were removed from home cages, and isolated for the duration of the test. All the tests were performed during daylight hours. The test apparatus was a 32-cm semi-transparent tube with adjustable internal diameter to prevent crossing over or turning of animals. Test animals were placed opposite ends of the tube and released simultaneously. Winners and losers were scored as one animal retreated from the tube completely. Tests in which the animal remained in the tube (five minutes maximum) were scored as draw. Inter-litter rank was determined by the number of winning within littermates (most win to least win = from 1, 2, 3, etc.).

## Histology and Immunohistochemistry

X-Gal staining and immunohistochemistry were performed following standard protocols described previously [17]. All tissues were fixed in 4% paraformaldehyde and embedded in Optimal Temperature Compound (OTC) after cryoprotection in 20–30% sucrose. 12 µm cryosections were used for X-gal staining and co-immunofluorescences. The Pw1nLacZ reporter gene expression faithfully identified the *Pw1* expressing cells [17], therefore, mouse anti-β-galactocidase antibody (1:500, Z3781: Promega) was used to represent *Pw1* expression in combination with either rabbit anti-AR (1: 1000, Santa-cruz, sc-816), anti-ERα (1:1000, Santa-cruz, sc-542), or AVP (1:1000, AB1565: Millipore) in co-localization study. For brain structure analysis, mice were anesthetized with Ketamine/Xylazine and transcardially perfused with 4% paraformaldehyde. Brains were post-fixed overnight and 30µm serial sections were obtained from the entire brains using a vibratome. Comparable regions consist of 13–16 coronal cross-sections. The brain section images were collected from the same brain coordinates (Bregma: between -0.70 to -0.94 mm). Typically the AVP⁺ PVN structure spanning in four sections were subjected for analysis. Floating sections were incubated with rabbit anti-AVP (1, 1000,

AB1565: Millipore) and detected using Vectastain ABC HRP kit (Vector Labs). Number of AVP$^+$ cells and the total area were quantified using ImageJ software [95].

## Statistical analysis

All figures and statistical analyses were generated using Prism 5 (GraphPad). Comparison between genotypes and sexes was performed using two-way ANOVA with Tukey's multiple comparison test. For male specific variables, Mann-Whitney U-test was applied for independent measurement. Correlation between two variables was tested by a simple linear regression analysis. $P$ value $< 0.05$ was considered statistically significant.

## Supporting information

**S1 Fig. Insulin action in *Pw1$^{+/pat-}$* adult males.** (**A**) Positive correlation of insulin levels with lean mass (top) and with body fat (bottom) was observed at at 3 months and 6 months of age, respectively. Correlation was determined by simple linear regression analysis. (**B**) Pancreatic insulin content and blood glucose levels at 10 months of age. Pancreatic insulin content was measured using acid-ethanol extraction protocol, followed by the insulin ELISA and normalized with total protein content. Columns, mean; bars, SEM; **, P< 0.01 by Mann-Whitney U test. (**C**) Typical images of Oil Red-O stained 8-month-old livers (top) and the quantification of lipid droplet number and size distribution (bottom), illustrating the differences in lipid content between the groups. P<0.001 by two-way ANOVA with Tukey's multiple comparisons. Original magnifications: x200.
(TIF)

**S2 Fig. *Pw1* colocalisation with sex hormone receptors.** *Pw1* reporter expression predominantly co-localizing with the nuclear expression of sex hormone receptors in diverse cell types. (**A**) pancreatic islets, (**B**) mono- and dinucleated hepatocytes, (**C**) anterior pituitary cells, (**D**) adipocytes, and (**E**) skeletal muscle.
(TIF)

**S3 Fig. Reduced aggressive behavior and social rank of *Pw1$^{+/pat-}$* adult males.** (**A**) Spontaneous fights among group-housed littermates were monitored per cage and individuals involved in fights were identified. ***, P< 0.001 by Fisher's exact test. (**B**) Result of social confrontation tube test in *Pw1$^{+/+}$* (+/+) and *Pw1$^{+/pat-}$* (pat$^-$) littermates from six litters. *Pw1$^{+/pat-}$* males (n = 10) and *Pw1$^{+/+}$* littermates (n = 13) were subjected for confrontation against each other and their i) intra-litter rank and ii) winning rate against unfamiliar opponents was determined for each animal. Squared in red on the diagonal line show matches within littermates. Mice were derived from C57B6 x *Pw1$^{+/p-}$* breeding except for Litter 1, which was derived from C57B6 x *Pw1$^{+/+}$* breeding). (**C**) Coronal sections of Pw1$^{IRESnLacZ}$ transgenic brain at 2.5 months of age revealing *Pw1* reporter gene expression (X-gal staining) in sexually dimorphic brain regions. BnST, the bed nucleus of the stria terminalis; mPOA, medial preoptic area; PVN, paraventricular nucleus of hypothalamus; MeA, medial amygdala; PIR, piriform cortex (x40). (**D**) Size comparisons of AVP$^+$ PVN area in four *Pw1$^{+/+}$* male siblings from C57B6 x *Pw1$^{+/+}$* breeding (Litter 1). Positive correlation was found between winning rate and AVP$^+$ cells.
(TIF)

## Acknowledgments

We thank Drs. AL. Denizot, RM. Correra, D. Ollitrault, K. Kyrylkova, S. Kyryachenko and JR. Courbard for scientific discussions, and Ms I. Lopez for technical assistance. We thank Ms. A.

Lacombe and Dr. T. Huby in the PreclinICAN, Institute of Cardiometabolism and Nutrition (IHU-ICAN, ANR-10-IAHU-05) for providing facility and technical advice for the metabolic analysis. We also thank the animal facility of Pierre and Marie Curie University for their excellent management and maintenance of laboratory animals.

## Author Contributions

**Conceptualization:** Karo Tanaka, Franck Oury, Giovanna Marazzi, David A. Sassoon.

**Data curation:** Karo Tanaka, Manon Rivagorda, Franck Oury, Giovanna Marazzi, David A. Sassoon.

**Formal analysis:** Karo Tanaka, Manon Rivagorda, Franck Oury, Giovanna Marazzi, David A. Sassoon.

**Funding acquisition:** David A. Sassoon.

**Investigation:** Karo Tanaka, Vanessa Besson, Manon Rivagorda, David A. Sassoon.

**Methodology:** Karo Tanaka, Vanessa Besson, Manon Rivagorda, David A. Sassoon.

**Project administration:** David A. Sassoon.

**Resources:** Vanessa Besson, Manon Rivagorda, Franck Oury, David A. Sassoon.

**Supervision:** Franck Oury, Giovanna Marazzi, David A. Sassoon.

**Validation:** Giovanna Marazzi, David A. Sassoon.

**Writing – original draft:** Karo Tanaka, Franck Oury, Giovanna Marazzi, David A. Sassoon.

**Writing – review & editing:** Karo Tanaka, Franck Oury, Giovanna Marazzi, David A. Sassoon.

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
