## [Decision Letter · Decision Letter 0]

15 Jun 2021

Dear Dr SASSOON,

Thank you very much for submitting your Research Article entitled 'Paternally expressed gene 3 (Pw1/Peg3) promotes sexual dimorphism in metabolism and behavior' to PLOS Genetics.

The manuscript was fully evaluated at the editorial level and by independent peer reviewers. The reviewers appreciated the attention to an important problem, but raised some substantial concerns about the current manuscript. Based on the reviews, we will not be able to accept this version of the manuscript, but we would be willing to review a much-revised version. We cannot, of course, promise publication at that time.

All three reviewers feel that the statistical analysis of the data is not appropriate and would need to be addressed throughout the manuscript. The use of appropriate statistical tests are likely to change the results and therefore the conclusions that can be drawn from the data presented. There are also some data apparently missing from the submission. The conclusion that 'Pw1 deficient males develop a gender-specific global shift of body metabolism towards accelerated adiposity, diabetic-like insulin resistance, and fatty liver in adult life' has been contested as not a conclusion that can be drawn from the data that have been presented and the idea that there is sexual dimorphism in lean growth phenotype of Pat animals is not supported here. A re-analysis of the data would be necessary to provide a basis for a new manuscript to be considered as well as attention to the way the data are presented/organised and how the Figures are constructed to illustrate those data.

If you decide to revise the manuscript for further consideration at PLOS Genetics, please aim to resubmit within the next 60 days, unless it will take extra time to address the concerns of the reviewers, in which case we would appreciate an expected resubmission date by email to plosgenetics@plos.org.

[LINK]

We are sorry that we cannot be more positive about your manuscript at this stage. Please do not hesitate to contact us if you have any concerns or questions.

Yours sincerely,

Rebecca J Oakey

Guest Editor

PLOS Genetics

John Greally

Section Editor: Epigenetics

PLOS Genetics

Reviewer's Responses to Questions

**Comments to the Authors:**

Reviewer #1: This submission reports on the characterisation of behavior and metabolism in a mouse model in which the imprinted Pw1/Peg3 gene has been ablated. Previous studies on this model and a previous independent targeting of this gene identified both metabolic and behavioral changes, albeit with some differences in findings in the different models/strain backgrounds as highlighted by the authors in their introduction. A key conclusion of this new study is that loss of Pw1/Peg3 results in gender-dependent metabolic and behavioural alterations whereby male mutant exhibit a “reduced masculinization of body composition, followed by a global shift of metabolism towards accelerated progression of obesity and related metabolic syndromes”.

Overall, this is an important finding which will be of interest to the wider genomic imprinting community and also those researching the sexually dimorphic consequence of early life adversities whereby, at least in some domains, males appear relatively more impacted than females.

A major comment on the manuscript is on the statistical approach. The authors have used paired comparisons but they should use ANOVA, with sex and genotype as factors. By using ANOVA, the authors will be able to make statistically supported conclusions regarding sex effects. For example, in Figure 1A this approach will likely show no effect of SEX and no interaction between SEX and GENOTYPE for body weight and loss of Peg3/Pw1 reduces body weight in both males and females. However, in terms of fat mass (Figure 1E & F), it is likely that an ANOVA would show an interaction between SEX and GENOTYPE, whereby loss of Peg3/Pw1 has little effect in females, but makes males more like WT females. Once this is done, the authors will need to update their abstract ect.

As an additional point, the authors keep switching comparisons between separating males and females, and Peg3/Pw1 and WT (e.g. Fig 1 D v E; Fig 2 C v G). They should be consistent in their comparisons which could also be resolved with the suggested statistical approach.

Tube test

There is some confusion regarding which animals are used in the tests to generate scores. The authors mention the use of “adult offspring derived from either Pw1+/+ or Pw1+/pat- males mated with C57/B6 WT females” and Pw1+/+ and Pw1+/pat littermates". Are the authors testing WT and mutant littermates against WT from a fully WT litter bred from Pw1+/+ males or, as indicated in S3, within litter bouts? It is also not clear what is meant by “found that the male progeny derived from Pw1+/pat- males were statistically less aggressive as compared to the ones derived from Pw1+/+ males?” The sentence implies that both mutant and WT males generated from Pw1+/pat- males are less aggressive than WT males from Pw1+/+ males? But there is no data on animals derived from Pw1+/+ males. A true test of social dominance takes into account rank within home cages which means animals must be tested against littermates, and rank order determined. Sentences should be modified in titles and elsewhere to reflect this.

Another point is that litter 1 is all WT and litter 4 has 2 mutant Peg3 males and no WT. If data from these two litters are excluded, are the findings still significant? Ideally for this test, litters should have mixed genotype in order for rank to be established in the context of genotype.

A third point is that the authors report that animals were initially housed together and then socially isolated before testing, as was done for the Garfield paper cited. There is a second paper on this model where the social group was maintained with no evidence of a social dominance phenotype (Rienecker et al., GBB 2020). This section and the M&M needs to be more clearly written and the conclusions reconsidered.

In the introduction, it would be helpful if the authors provided more explicit details on what behavioural and metabolic phenotypes have been reported in response to loss of Peg3 in their and other models. For example, has the timing of sexual maturity in males and females been reported? What precise behavioural phenotypes have been reported? Similarly, the authors should comment on previous reports of sexually dimorphic outcomes in response to loss of Peg3/Pw1 (Kim et al., Plos One 2013; Tunster et al., Front Dev Biol 2018). This would provide a more cohesive basis for the work undertaken and it’s interpretation.

Minor points

Introduction

Should “we characterized Pw1 deficient phenotypes in male and female mice at different postnatal development” read “we characterized Pw1 deficient phenotypes in male and female mice at different stages of postnatal development”

Results

“Consistent with changes in IGF-1 levels, we observed that gene expression levels of pituitary Gh, hepatic growth hormone receptor (Ghr) and Igf1 were reduced in Pw1+/pat- males as compared to Pw1+/+ littermates (Fig 2B).” change to “Consistent with changes in IGF-1 levels, we observed that gene expression levels of Gh in the pituitary gland and growth hormone receptor (Ghr) and Igf1 in the liver were reduced in Pw1+/pat- males as compared to Pw1+/+ littermates (Fig 2B).”

To avoid the use of “trends” suggest “Notably, these trends were inverted at 6-months of age that displayed higher insulin secretion and clearance in Pw1+/pat- males.” changes to “Notably, these patterns were inverted at 6-months of age that displayed higher insulin secretion and clearance in Pw1+/pat- males.”

“Prader-Willy syndrome (PWS)” change to “Prader-Willi syndrome (PWS)”

Reviewer #2: Review uploaded as an attachment.

Reviewer #3: Paternally expressed gene 3 (Pw1/Peg3) promotes sexual dimorphism in metabolism and behaviour.

Here the authors present interesting data regarding the function of the paternally-expressed imprinted gene Pw1/Peg3. While many experiments were performed to a good standard, I do not feel that the paper in its current form has sufficient novelty, explanatory power or statistical rigor to merit publication in PLoS genetics. Below I present my view of the weaknesses in the authors' argument, and some suggestions of how the work might be improved.

The authors state 3 key claims for this work.

Mice carrying paternally inherited Pw1/Peg3 allele manifest deficits in:

1) GH/IGF1 dependent growth before weaning.

2) Sex steroid-dependent masculinisation during puberty.

3) Insulin-dependent fat accumulation in adulthood.

Pw1/Peg3 paternal mutant mice (hereafter Pats) are born small and maintain their growth deficit into adult life, in both sexes.

• It is difficult to determine to what extent the growth retardation in Pat mice is due to early growth deficit, or the combination of this and reduced postnatal growth trajectory. The data in Fig1A should be also plotted as % wild-type (WT) weight, or and/growth rates should be calculated. In this way, deceleration of growth associated with Pw1/Peg3 deficit can be easily visualised and the timing pinpointed.

• In Fig1B body weight and glycemia are said to be correlated, yet no statistical value is stated. R or R-squared values as well as p-values should be reported.

• Food consumption is stated to be lower in mutant mice, yet this is expected if body mass is reduced. These data should be normalised to total body mass and lean body mass to determine if the mutant animals eat a disproportional amount, which would be indicative in a change in central metabolic homeostasis.

• The authors claim that because the Pat males resemble the WT females in size, there is therefore an absence of sexual dimorphism in the absence of Pw1/Peg3. This is faulty logic. In terms of the early life growth phenotype the males and females respond similarly to Pw1/Peg3 deletion (i.e., both are growth restricted to a similar extent, as is the body composition prior to 3 months). These data are obscured by the presentation in Figure 1D. These data should be shown as in E, with males and females on separate axes, allowing comparison between WT and Pat animals over time. Indeed, Fig1E clearly shows that both males and females have reduced lean mass to a similar degree.

• Further to this, the puberty-associated lean mass does not appear to be affected in the Pat males – compare the gradient of the lean mass increase in Fig1E between genotypes, it is identical. I would rather interpret this to indicate that the developmental deficit in Pw1/Peg3 has caused a reduction in muscle mass (as previously reported by this group and others), and that is maintained into adulthood.

• Statistics applied to repeated measures of body mass is inappropriate. Each animal at a different timepoint, nor fat and lean mass are independent variables, therefore pairwise statistics without multiple comparison correction are not appropriate. Consequently the authors should have performed a 2-Way ANOVA for each sex comparing the effects of time and genotype on lean and fat mass.

Link to IGF1/GH axis.

The authors report the interesting finding that IGF1 levels are reduced in 3-week old Pat mice. They attribute this to an alteration in the pituitary GH axis since in males pituitary Gh mRNA levels are reduced. This is an interesting finding, but there are some discrepancies.

• Growth retardation is evident from birth, yet in rodents GH does not drive hepatic IGF1 secretion until around the second postnatal week.

• The reduction in IGF1 is seen in both sexes (though the female data is underpowered) and is correlated with body size in both sexes, yet pituitary Gh and hepatic Igf1 mRNA levels are only different in males, what is the explanation for this?

• Regarding Figure 2C the authors state that there is sexually dimorphic behaviour regarding the male-specific rise in IGF1 post-puberty (which they say does not happen in Pat males). This way of showing the data distorts the picture. Clearly Pats of both sexes have an early IGF1 deficit (~200ng/mL vs ~350ng/mL) which is mostly normalised in the adults of both sexes. The data should be shown with sexes separated and WT and Pat on the same graph.

Metabolic phenotype

• Is the data in Fig2D from fasted or free-fed animals? The error bar appears to be missing for the 3mo pat females. What is the n for these data?

• Fig2G, the paper text says the animals are 6 months old, the figure legend say 10 mo, which is it?

• What is the n for Fig2H?

In conclusion, I do not feel that the data presented in this paper supports a conclusion that there is sexual dimorphism in lean growth phenotype of Pat animals. Therefore, the later association of this phenotype with masculinising hormones cannot be upheld. Moreover, while the observation of reduced IGF1 in Pats is interesting, it is present in both sexes and cannot be entirely explained by alteration of the pituitary GH axis. However, there is convincing evidence of male-specific adipose weight gain in the mutant animal in later life, and this is associated with the expected glucose intolerance, beta cell expansion and hyperglycaemia. While interesting, the authors are not able to provide an explanation for this at the molecular level. Moreover, such changes in body composition and metabolism have previously been reported by the Keverne/Curley group in an independent Pw1/Peg3 deletion model.

Behavioural phenotyping and localisation of PW1/PEG3 in the adult male brain.

• What statistics were performed on the behavioural data? This should be stated.

• Expression of PEG3 in the AVP neurons of the PVN has been previously reported: https://pubmed.ncbi.nlm.nih.gov/12399444/

• In Fig4A how many animals were assayed, this is not stated? Also, the stats is inappropriate, it should be analysed by One-Way ANOVA since the individuals at multiple timepoints are not independent. The error bar for the Pat at 9 weeks is missing.

• The ovelap between PW1/PEG3 expression and AR in the gonads is minimal – would this be sufficient to account for the reduction in testosterone observed in Fig4B?

• Would a transient reduction of testosterone observed be sufficient to account for the metabolic phenotype? The critical causal experiment would be to replace testosterone levels in mutant mice at this age, and see if the metabolic and behavioural phenotypes were reversed.

In conclusion, the observed transient reduction in testosterone levels, and association of PW1/PEG3 expression with sex hormone-responsive cell populations is a very interesting finding. However, at this stage the work appears preliminary with not much causal link to phenotype. A hormone replacement/conditional deletion experiment would be required to elevate this work.

**Have all data underlying the figures and results presented in the manuscript been provided?**

Reviewer #1: Yes

Reviewer #2: **No: **Numerical data underlying graphs or summary statistics have not been provided in spreadsheet form in supporting information.

Reviewer #3: Yes

PLOS authors have the option to publish the peer review history of their article (what does this mean?). If published, this will include your full peer review and any attached files.

Reviewer #1: No

Reviewer #2: No

Reviewer #3: No

---

## [Decision Letter · Decision Letter 1]

20 Nov 2021

Dear Dr SASSOON,

Thank you very much for submitting your Research Article entitled 'Paternally expressed gene 3 (Pw1/Peg3) promotes sexual dimorphism in metabolism and behavior' to PLOS Genetics.

The manuscript was fully evaluated at the editorial level and by independent peer reviewers. The reviewers appreciated the attention to an important topic but identified some concerns that we ask you address in a revised manuscript. These include a general attention to improve the language used and some specific issues around particular over-stretching of statements in the text from the data presented.  

We therefore ask you to modify the manuscript according to the review recommendations. Your revisions should address the specific points made by each reviewer.

[LINK]

Yours sincerely,

Rebecca J Oakey

Guest Editor

PLOS Genetics

John Greally

Section Editor: Epigenetics

PLOS Genetics

Reviewer's Responses to Questions

**Comments to the Authors:**

Reviewer #1: The authors have undertaken considerable work to address comments made by reviewers which significantly improve the understanding of the manuscript. As a minor point, while tests for aggression have predominantly been developed on male rodents, non-pregnant females can exhibit aggression (Oliveira 2021). Moreover, there is evidence of increased aggression in lactating Peg3 mutant females (Champagne 2009). The authors have not tested aggression in females in this study. The title of the section related to tube test should be amended.

From

Paternal Pw1 deficiency reduces male-specific aggressive behaviors and social dominance.behavior.

To

Paternal Pw1 deficiency reduces aggressive behaviors and social dominance.behavior in males.

Reviewer #2: The authors have done a good job responding to my concerns and the manuscript is significantly improved. I suggest that the manuscript requires 'minor revisions' only because there are a few errors throughout the text. Some examples include "Body weight at 7-week-old revealed..." (this should read "Body weight at 7 weeks of age revealed...") and "Sex differences emerges..." (this should read "Sex differences emerge..."). I suggest the authors go through the manuscript one more time to correct these and other grammatical and sentence structure issues.

**Have all data underlying the figures and results presented in the manuscript been provided?**

Reviewer #1: Yes

Reviewer #2: Yes

PLOS authors have the option to publish the peer review history of their article (what does this mean?). If published, this will include your full peer review and any attached files.

Reviewer #1: No

Reviewer #2: No

---

## [Editor Report · Decision Letter 2]

20 Dec 2021

Dear Dr SASSOON,

We are pleased to inform you that your manuscript entitled "Paternally expressed gene 3 (Pw1/Peg3) promotes sexual dimorphism in metabolism and behavior" has been editorially accepted for publication in PLOS Genetics. Congratulations!

Yours sincerely,

Rebecca J Oakey

Guest Editor

PLOS Genetics

John Greally

Section Editor: Epigenetics

PLOS Genetics

Comments from the reviewers (if applicable):

**Data Deposition**

http://datadryad.org/submit?journalID=pgenetics&manu=PGENETICS-D-21-00562R2

**Press Queries**

---

## [Editor Report · Acceptance letter]

10 Jan 2022

PGENETICS-D-21-00562R2 

Paternally expressed gene 3 (Pw1/Peg3) promotes sexual dimorphism in metabolism and behavior 

Dear Dr SASSOON, 

We are pleased to inform you that your manuscript entitled "Paternally expressed gene 3 (Pw1/Peg3) promotes sexual dimorphism in metabolism and behavior" has been formally accepted for publication in PLOS Genetics! Your manuscript is now with our production department and you will be notified of the publication date in due course.

With kind regards,

Olena Szabo

PLOS Genetics

On behalf of:
